# Pregnant Inuit Women’s Exposure to Metals and Association with Fetal Growth Outcomes: ACCEPT 2010–2015

**DOI:** 10.3390/ijerph16071171

**Published:** 2019-04-01

**Authors:** Per I. Bank-Nielsen, Manhai Long, Eva C. Bonefeld-Jørgensen

**Affiliations:** 1Centre for Arctic Health and Molecular Epidemiology, Department of Public Health, Aarhus University, 8000 Aarhus C, Denmark; pbn@ph.au.dk (P.I.B.-N.); ml@ph.au.dk (M.L.); 2Greenland Center for Health Research, University of Greenland, 3900 Nuuk, Greenland

**Keywords:** Greenland, Arctic, heavy metals, perinatal risks, smoking, reproductive health, environmental pollutants

## Abstract

Environmental contaminants such as heavy metals are transported to the Arctic regions via atmospheric and ocean currents and enter the Arctic food web. Exposure is an important risk factor for health and can lead to increased risk of a variety of diseases. This study investigated the association between pregnant women’s levels of heavy and essential metals and the birth outcomes of the newborn child. This cross-sectional study is part of the ACCEPT birth cohort (Adaption to Climate Change, Environmental Pollution, and dietary Transition) and included 509 pregnant Inuit women ≥18 years of age. Data were collected in five Greenlandic regions during 2010–2015. Population characteristics and birth outcomes were obtained from medical records and midwives, respectively, and blood samples were analyzed for 13 metals. Statistical analysis included one-way ANOVA, Spearman’s rho, and multiple linear and logistic regression analyses. The proportion of current smokers was 35.8%. The levels of cadmium, chromium, and nickel were higher compared to reported normal ranges. Significant regional differences were observed for several metals, smoking, and parity. Cadmium and copper were significantly inversely related to birth outcomes. Heavy metals in maternal blood can adversely influence fetal development and growth in a dose–response relationship. Diet and lifestyle factors are important sources of toxic heavy metals and deviant levels of essential metals. The high frequency of smokers in early pregnancy is of concern, and prenatal exposure to heavy metals and other environmental contaminants in the Greenlandic Inuit needs further research.

## 1. Introduction

Environmental contaminants such as toxic heavy metals and persistent organic pollutants (POPs) are transported to the Arctic through long-range atmospheric and ocean currents [1,2,3,4,5,6,7,8]. Heavy metals (e.g., mercury, lead, and cadmium) and POPs are introduced to the ecosystems, where they bio-accumulate and bio-magnify up through the food chain in, e.g., fish and marine mammals [3,6,9,10,11,12,13,14,15].

Heavy metals is a general term that applies to metals and metalloids, which have a relatively high density and are considered to be toxic to living organisms and the environment in certain concentrations [9,10,11].

Metals and other elements occur naturally in the Earth’s crust. However, human and environmental exposure has increased with increasing use of heavy metals in industry, agriculture, technology, and medicine, as well as from domestic settings and workplaces [9,10,11].

Some metals and trace elements are essential nutrients for biochemical processes in the human organism. Insufficient or otherwise deviant blood concentrations of these essential metals and trace elements may result in various diseases [9,10,11,16,17]. Because of their relatively high toxicity, mercury (Hg), lead (Pb), arsenic (As), cadmium (Cd), and chromium (Cr) rank among chemicals that are of major public health concern by the World Health Organization (WHO) [16,17,18]. Even at lower levels of exposure, these metallic elements are considered systemic toxicants that are known to affect and damage multiple organ systems [9,10,11]. They are also classified as “known” or “probable” human carcinogens according to the US Environmental Protection Agency and the International Agency for Research on Cancer [11,19].

The severity of toxicity as well as organ systems affected depends on the heavy metal type, chemical form, whether it is single or mixtures of compounds, exposure dose and time, exposure route, physiologically based pharmacokinetic model, and the age of the exposed organisms, including humans [9,10,11,20,21,22].

Exposure to heavy metals may lead to increased risks of cancer, allergies, neurological diseases, decreased cognition, and endocrine disorders [23,24,25,26,27]. The latter may adversely affect the hormonal system and disrupt the reproductive and/or immune system of both the exposed adult and pregnant women and their offspring, causing birth defects or developmental disorders [23,24,25,26,27]. Pregnant women are therefore particularly vulnerable, as exposure may affect fetal development and, thus, the next generation [23,25,26,27,28,29]. Some contaminants are transferred from the mother to the fetus through the placental barrier and to the newborn through breastfeeding [10,11,15,30,31,32,33].

The developmental period of the fetus and the newborn is the most sensitive period in life, making them particularly vulnerable to exposure of heavy metals [23,25,33]. Therefore, heavy metals are important risk factors to be investigated and monitored for assessment of the next generation’s health [16,17,23,25]. Factors that can elucidate pregnant women’s exposure and potential effects on fetal development are birth outcomes such as weight, length, head circumference, gestational week, and, to some degree, the APGAR score at five minutes after birth (APGAR5) which assesses physical condition of the newborn and the need for intervention.

The traditional diet of the Greenlandic Inuit relies on marine food, such as seal, whale, and polar bears at the top of the food chain, and furthermore fish, and therefore, they are exposed to relatively higher levels of heavy metals [3,6,12,28,34]. It has been suggested that there are regional differences in the intake of marine mammals and the exposure to certain types of heavy metals and POPs [6,12,23,34,35,36].

There are reports on a decreasing trend in blood levels of contaminants for some of the Arctic populations, indicating regulation through legislation as well as health interventions are working [12,36,37]. However, the problems with these environmental contaminants are not yet solved because of their persistency. Humans are still being exposed to heavy metals as well as POPs [30,37,38,39]. The aim of this study was to investigate the association of Greenlandic pregnant Inuit women’s levels of heavy metals and essential metals and the birth outcomes.

## 2. Materials and Methods

### 2.1. Study Population

This cross-sectional study is nested in the birth cohort ACCEPT (Adaption to Climate Change, Environmental Pollution, and dietary Transition) [40]. The overall aims of the ACCEPT project was to establish a geographical Greenlandic mother–child cohort, including establishment of a biobank to evaluate the impact of lifestyle, reproductive factors, diet, and other health outcomes, as well as environmental contaminants on birth outcomes and follow-up on child health and development [40].

Data were collected from 2010–2015 in five regions of Greenland: North (RN), Disko Bay (DB), West (W), South (S), and East (E) (Figure 1).

The criteria for inclusion were pregnant Inuit women ≥18 years of age who had lived more than 50% of their lives in Greenland.

ACCEPT enrolled 614 pregnant women. Among them, 42 participants had not lived in Greenland for more than 50% of their lives, five participants were younger than 18 years of age, 33 women gave no information of life duration in Greenland, and 19 participants withdrew their consent to participate or had an early abortion and were excluded. Then, 509 women fulfilled the criteria for inclusion; however, of these, 12 participants had abortions or miscarriages, five participants had no birth outcome information, four twin pairs and two stillborn were excluded, and finally, 487 live born singletons were included in our study analysis (Figure 2).

A Danish- and Greenlandic-speaking medical doctor recruited the participants in 2010–2011 and Greenlandic midwives and the study research team in 2013–2015. Enrolment and inclusion to the ACCEPT study have previously been reported [23,27,28,34]. Data on the women’s height (in meters), weight (in kilograms), smoking, and alcohol intake were obtained from medical records. Birth outcomes, including birth weight and birth length, head circumference, gestational age, and APGAR5 scores at birth, were obtained from the Greenlandic Country Doctors office.

Samples of venous blood were collected from the pregnant women at inclusion to the study and stored at −80 °C until analysis.

Prior to data collection, all participants received detailed descriptions of the objective of the project. Informed consent was obtained from all participants and data were anonymized after collection. In addition, participants were informed that participation was voluntary, and they could withdraw their consent at any time.

This study was approved by the Ethical Committee for Scientific Investigations in Greenland (J.no. 011-0035-09/gope) as well as the Danish Data Protection Agency (J.no. 2011-41-6367) and conducted in accordance with the Helsinki II Declaration.

### 2.2. Determination of Blood Components

#### 2.2.1. Blood Metals

Thirteen blood metals, including heavy metals mercury (Hg), lead (Pb), arsenic (As) and cadmium (Cd), chromium (Cr), manganese (Mn), and nickel (Ni), as well as essential metals, including selenium (Se), iron (Fe), copper (Cu), zinc (Zn), magnesium (Mg), and calcium (Ca), were measured using inductively coupled plasma mass spectrometry (ICP-MS) at the accredited element laboratory, Institute for Bioscience—Arctic Research Centre, Aarhus University, Denmark. The quality was ensured by repeated analyses and frequent analysis of certified reference material (Seronorm™), as well as by participation in the Quality Assurance of Information in Marine Environmental monitoring (QUASIMEME), an inter-laboratory comparison program [41]. Unless otherwise stated, whole blood values are given. The included metals of this study with reference to reported normal ranges for non-Inuit, nonpregnant, and function in humans are given in Appendix A.

#### 2.2.2. Plasma Cotinine

Cotinine is a metabolite of nicotine and is therefore proportional to the amount of exposure to tobacco smoke and, thus, serves as a biomarker for recent exposure to tobacco smoke. The Calbiotech Cotinine Direct ELISA Kit from Calbiotech Inc., CA, USA was used to measure plasma cotinine at the Centre for Arctic Health and Molecular Epidemiology, Department of Public Health, Aarhus University, Denmark.

#### 2.2.3. Plasma Fatty Acids

The ratio between n-3 polyunsaturated fatty acids and n-6 fatty acids (n-3/n-6 ratio) is known to be a strong indicator for the consumption of marine food items vs. imported food items [23]. Plasma fatty acids were determined by capillary gas–liquid chromatography at the Biology Department, University of Guelph, Canada [23].

### 2.3. Birth Outcomes

The birth outcomes are basic measurements of the newborns [42]: Birth weight (grams), birth length (cm), head circumference (cm), gestational age (weeks) at birth, and APGAR5 scores.

Categorized outcomes have the following definitions:Low birth weight: Defined by the WHO as weight at birth less than 2500 g [43].Preterm birth: Defined by the WHO as babies born before 37 completed weeks of gestation [44].Low APGAR5: Defined as a score of less than 7 [45].

The APGAR score assesses the newborns and the need for intervention to establish breathing. The score is reported at one minute, five minutes and at five-minute intervals thereafter until 20 min for infants with scores below 7 [45]. The method was devised by Dr. Virginia Apgar in 1952 and consists of the assessment of: Color, heart rate, reflexes, muscle tone, and respiration. Each of the five components is given a score of 0, 1 or 2 and, thus, a maximum score of 10 [45].

### 2.4. Statistical Analysis

Data were analyzed with IBM SPSS Statistics version 20.0 (International Business Machines Corp., Armonk, NY, USA) with a significance level of 5%, and 95% confidence intervals (95%CI) were calculated. Unless otherwise stated, only significant results are given in the text.

Data distribution was checked by Q–Q plot and natural logarithmic transformed data made the distribution more symmetrical and improved the normality of data.

One-way ANOVA was applied to compare the levels of metals in pregnant women from different Greenlandic regions. When ANOVA showed statistical significance, complementary multiple comparison post hoc tests were performed. A test for equal variances of variables was performed using Levene’s test. Fisher’s least-significant difference test was used if equal variance was observed; otherwise, Dunett’s T3 was used. One-way ANOVA was applied to compare birth outcomes for the quartiles of blood metals. Pearson’s chi-square was applied to check the regional difference of categorical variables. Student’s t-test was used to compare the levels of cotinine in smokers/nonsmokers, respectively. Spearman’s rho was applied to assess the bivariate correlation of the metals for all regions in total as well as stratified by region, and also to assess the correlation between metals and lifestyle factors.

The association for the maternal levels of blood metals and the continuous birth outcome data was analyzed by multiple linear regression analysis and was stratified by gender. For the association between maternal blood metal concentrations and the categorical birth outcome data (low birth weight, preterm birth, low APGAR score), logistic regression analysis was used. We built the models one element by one. The identification of potential confounders was based on literature knowledge and directed acyclic graphs (DAGs) supported by the “Change in estimate” method [46,47]. For the “Change in estimate”, estimate changes >10% were used. Data were analyzed as raw data, core adjustment for age, body mass index (BMI), alcohol during pregnancy, cotinine, parity and n-3/n-6 ratio, and then further adjusted for gestational age and region, respectively.

## 3. Results

### 3.1. Population Characteristics

Table 1 presents the characteristics of the study population.

The study population included 509 pregnant women (18–43 years of age) with a median age of 27.0 years and median pre-pregnancy BMI of 24.3 kg/m^2^. Primary school, secondary, and university education were 29.4%, 47.1%, and 23.5%, respectively, and 56.8% of the participants lived in the West region. More than a third (35.8%) of the women were current smokers at the beginning of their pregnancy. The proportions of smokers were statistically significant different among the regions with the highest frequency in East in the order: East (E) > South (S) > North (RN) > West (W) > Disko Bay (DB). No significant difference was found for plasma cotinine among the regions for smokers or nonsmokers; however, current smokers had significant higher plasma cotinine levels compared to nonsmokers (median: 63.7 µg/L vs. 0.5 µg/L, *p* < 0.001).

For alcohol consumption during pregnancy, a borderline significant difference (*p* = 0.08) among the regions was found, possibly explained by only one person’s (from the North region) consumption of >2–3 times a month.A significant difference was found among the regions for parity, defined as a full-term pregnancy, with a higher median in the East region. No significant difference was found when parity was categorized into 0, 1–2 or ≥3 prior full-term pregnancies. The n-3/n-6 ratio was not significantly different among regions, although a slightly higher median value in the East region was observed.

### 3.2. Blood Levels of Metals

Table 2 presents the pregnant women’s blood levels of metals. Significant regional differences were observed for Hg, As, Cd, Cr, Se, Fe, Cu, Zn, Mg, and Ca. The median level of Hg in the blood of the women was 4.2 µg/L, and there was a difference between the regions as follows: E > RN > DB = S > W. The arsenic median level was 4.2 µg/L, with a difference between the regions as follows: DB > W = RN = S > E. The median level of Fe was 436.8 mg/L, with a difference between the regions as follows: S > RN > W > DB > E. For Zn, the median level was 4.6 mg/L, with a difference between the regions as follows: S = RN > W > DB = E. The median level of Mg was 37.6 mg/L, and there was a difference between the regions as follows: E > W = S > RN > DB. No significant differences across regions were found for Pb, Mn, Ni, and Plasma-Se.

### 3.3. The Correlations of Metals and Lifestyle

Table 3 presents Spearman’s correlation coefficient between metals, age, and lifestyle factors.

Hg was positively correlated with the n-3/n-6 ratio; Pb positively correlated with age, cotinine, and n-3/n-6 ratio; As positively correlated with age while negatively correlating with smoking; Cd positively correlated with both smoking and cotinine. Cr negatively correlated with smoking and the n-3/n-6 ratio; Mn correlated negatively with the n-3/n-6 ratio. Ni positively correlated with the n-3/n-6 ratio. Se and plasma-Se were positively correlated with age and the n-3/n-6 ratio. Fe positively correlated with age but significantly negatively correlated with smoking and cotinine level. Cu positively correlated with smoking. Zn negatively correlated with smoking and cotinine level. Ca positively correlated with smoking and cotinine but negatively with age.

Moreover, we found a negative correlation between age and smoking (r = −0.125; *p* < 0.01) and cotinine (r = −0.171; *p* < 0.01), respectively. Smoking positively correlated with cotinine (r = 0.654; *p* < 0.01). Age positively correlated with the n-3/n-6 ratio (r = 0.171; *p* < 0.01) (not shown).

Significant correlations between several metals were found (Appendix A). Hg was positively correlated to Se, plasma-Se, Pb, As, Cd, and Fe. Moreover, Fe was positively correlated to Hg (r = 0.118; *p* = 0.01), Se (r = 0.177; *p* < 0.01), Pb (r = 0.115; *p* < 0.01), Cd (r = 0.138; *p* < 0.01), Cr (r = 0.098; *p* = 0.03), and Mn (r = 0.112; *p* = 0.01) in the order of Se > Cd > Hg > Pb > Mn > P-Se > Cr but negatively to Cu (r = −0.134; *p* < 0.01); negative correlations were observed between plasma-Se and Cd (r = −0.119; *p* < 0.01), Cr (r = −0.162; *p* < 0.01) and Mn (r = −0.154; *p* < 0.01), and Cd and Cr (r = −0.226, *p* < 0.01). (Appendix A and not shown).

### 3.4. Birth Outcomes

Table 4 presents the birth outcomes. No statistically significant differences between regions were found for the selected birth outcomes: Weight, length, head circumference, gestational age, and APGAR5. The median birth weight was 3615.5g. Median birth length and head circumference were 51.0 cm and 35.0 cm, respectively. The median gestational age was 39 weeks. APGAR5 scores >7 were observed for 97.4% of the offspring. Across the regions, between 2.2–3.2% of the newborns were assigned APGAR5 scores of <7, except North and East, which had no children in this category.

### 3.5. Association between Prenatal Exposure to Metals and Birth Outcomes: Continuous Data

Table 5 presents the association between prenatal metal exposure and birth weight for the continuous data. The raw data for Pb were borderline inversely related to birth weight (*p* = 0.076). The maternal blood level of Cd was negatively associated with birth weight both as raw data as well as upon adjustments. For Cu, the raw data were negatively associated with birth weight and upon core and region adjustment, whereas the inverse association for Ca disappeared upon further adjustments (Table 5).

The associations between prenatal exposure to metals and birth length are given in Appendix A. Estimates for Cd were negatively associated with birth length both for the raw data (β = −0.241; 95%CI: −0.481; −0.001; *p* = 0.049), upon core adjustment (β = −0.308; 95%CI: −0.556; −0.061; *p* = 0.015), core adjustment plus gestational age at birth (β = −0.228; 95%CI: −0.450; −0.007; *p* = 0.043) and adjustment for core plus region (β = −0.303; 95%CI: −0.549; −0.057; *p* = 0.016) (Appendix A).

A statistically significant association was found between head circumference and raw data of Hg (β = −0.023; 95%CI: −0.046; −0.0003; *p* = 0.047) and Zn adjusted for core and gestation age (β = −0.00016; 95%CI: −0.00031; −0.000006; *p* = 0.038) (Appendix A).

The associations between prenatal exposure to metals and gestational age at birth are given in Appendix A. Maternal blood levels of Cu were negatively associated with gestational age both for raw data (β = −0.0006; 95%CI: −0.0011; −0.00007; *p* = 0.026), upon core adjustment (β = −0.001; 95%CI: −0.001; −0.00006; *p* = 0.048) and adjustment for core plus region (β = −0.001; 95%CI: −0.001; −0.00002; *p* = 0.042). Raw data Zn (β = 0.0003; 95%CI: 0.0001; 0.0005; *p* = 0.026) and Mg (β = 5.8 × 10^−5^; 95%CI: 0.00001; 0.0001; *p* = 0.011) were weakly positive, while Ca were negatively associated with gestational age at birth (β = −0.00003; 95% CI: −0.00004; −0.00001; *p* = 0.004), but upon adjustments, the associations disappeared (Appendix A).

When stratified by gender, these associations showed clear differences in blood levels of metals (Appendix A). Pb, Ni, and Cr were negatively and positively related to female birth weight. After adjustments, Ni was negatively associated with head circumference and gestational age for girls. For boys, no statistically significant association between Cd and birth weight and birth length was observed (Appendix A).

### 3.6. Quartiles of Birth Outcomes and Metal Concentration

Appendix A presents the birth outcomes divided into quartiles of heavy metal and essential metal concentration, respectively. Several significant dose–response relationships were found within the birth outcomes by comparing the differences between quartiles of metal concentration.

Lower birth weight and smaller head circumference was observed for the highest quartile of Pb. Birth weight and birth length was dose-dependently negatively related to Cd (Appendix A) and Cu (Appendix A).

The gestation age was negatively related to As and Cd exposure (Appendix Aa). The quartiles of APGAR5 were found to be borderline positively related to Cd (Appendix A) and Cu (Appendix A) but negatively to Cr (Appendix A).

### 3.7. Odds Ratio: Association between Prenatal Exposure to Metals and Risk of Low Birth Weight, Low APGAR5, and Preterm Birth

Table 6a presents the association between prenatal exposure to metals and risk of low birth weight (<2500 g). Blood levels of Cd increased the risk of low birth weight with adjusted ORs in the range of 1.425–1.517 (Table 6a). Positive association between Ni and low birth weight risk were observed both for the raw data and core adjustments but disappeared upon further adjustment for gestational age and region. For Fe and Mg, the raw data were negatively associated to the risk of low birth weight, but the significance disappeared upon adjustments.

Table 6b presents the association between prenatal exposure to metals and risk of low APGAR5 (APGAR score <7). All OR estimates for Cd were significantly below 1.

Table 6c presents the risk of prenatal exposure to metals and preterm birth (<37th week). Positive association of Cd and preterm birth was observed in raw data, whereas upon adjustments, the association attenuated. Although weakly, the risk of preterm birth was significantly positively associated to Cu, both for the raw data and upon adjustments. A weakly significant association (ORs were close to 1) between Zn, Mg, and Ca with preterm birth was observed for the raw data but disappeared upon adjustments.

## 4. Discussion

From 2010–2015, data were collected on blood levels of 13 metals from 509 pregnant women with a median age of 27 years living in one of the following Greenlandic regions: North, Disko Bay, West, South or East for more than 50% of their lives.

The present study included the populations of the ACCEPT sub-studies from 2010–2011 and 2013–2015 [28,34].

Since no normal range data were found for the Inuit, we compared our metal data with reported normal range data for healthy, nonpregnant, non-Inuit populations [48,49].

The median levels of heavy metals such as mercury (Hg) and lead (Pb) were relatively lower than in previously reported data for the Greenlandic population, indicating a decline over time [23,39,50]. Median blood levels of Cd, Cr, and Ni were above the reported clinical normal ranges for nonpregnant, non-Inuit populations [48,49].

In some regions, especially North and East, deviant levels of both essential metals and heavy metals and lifestyle factors were observed.

Significant inverse relations were observed for birth weight and maternal levels of Pb, Cd, and Cu; birth length and maternal levels of Cd and Cu; head circumference and maternal levels of Pb and gestational age and maternal levels of Cu, As, and Cd.

### 4.1. Population and Lifestyle Characteristics

The population characteristics across regions were similar in general, but significant regional differences were found for smoking, alcohol consumption during pregnancy, and parity.

#### 4.1.1. Smoking and Cotinine

A total of 35.8% of the participants were current smokers in their early pregnancy—we observed a decline compared to previous Greenlandic birth cohorts like the Disko Bay Birth Cohort (60%), INUENDO (56%), and IVAAQ (45%) [28,34]. However, the East region had 68% smokers among the pregnant women.

Although the smoking rate seems to have declined in some regions compared to previous reports, the high smoking rate is a major public health challenge in Greenland because of its high prevalence and related morbidity and mortality from a number of diseases. Lung cancer is the most common cancer in Greenland, amounting to 34% of cancer deaths during 2000–2014 [51].

Smoking is a well-known source of heavy metals, affects growth and development of the fetus, and increases the risk of low birth weight and preterm birth and, thus, health risk in childhood and later in life [52,53]. Collection of the smoking data and issues related to this matter are described in previous ACCEPT sub-studies [28,34].

Significant differences in current smoking frequency were observed across the regions, being the highest in East (72.2%), followed by the South (40.9%), then the North (40.6%), then the West region (33.7%), and Disko Bay (32.2%). There was a significant difference in plasma cotinine level between smokers and nonsmokers (*p* < 0.001) but not across the regions. Cotinine is a proportional marker for recent tobacco exposure and can support the smoking questionnaire that may be subject to reporting and/or recall bias [52].

#### 4.1.2. Alcohol and Parity

Alcohol consumption during pregnancy was borderline different between regions, although only one participant from North region accounted for the difference and consumed alcohol ≥2–3 times a month.

Several studies found prenatal alcohol exposure as a risk factor for reduced birth weight and length, head circumference, and physical and mental development [34,54,55]. One study found that low exposure was not associated with low birth weight, preterm birth or intrauterine growth restrictions [56].

For parity, a significant difference between regions was found, especially within the East region, standing out with a higher median for birth. For the East region, 76.4% of the women had one or more full-term pregnancies.

#### 4.1.3. Body Mass Index and n-3/n-6 Ratio

BMI was not significantly different between regions, but a higher BMI was observed in the East region. Whereas the other regions were within the range of a normal BMI, East was above the WHO cut-off for overweight of ≥25 kg/m^2^ [57]. Defining overweight according to this WHO international standard might overestimate the number of individuals with elevated BMI in Greenland, since it is derived from predominantly Caucasian populations and, thus, a different ethnicity, different body proportions and distribution of, e.g., adipose tissue [58,59,60]. For the present study, the establishment of an Inuit-specific cut-off of, e.g., 27.5 kg/m^2^ would classify no region as overweight when examining the median BMI and may be beneficial for monitoring public health in Greenland [58,59].

No significant regional difference was found for the n-3/n-6 ratio, but the highest levels were observed in the East, indicating a higher consumption of a marine diet in this region.

### 4.2. Blood Levels of Heavy Metals

#### 4.2.1. Mercury and Lead

The blood Hg levels were significantly different across regions and above reported clinical normal range exclusively in the East region (Appendix A). Compared to the Canadian Inuit (median 3.51 µg/L [36]) and North Norwegian populations (median 1.51; 1.45; 1.2 µg/L [29,61,62]), the blood Hg levels in the pregnant women in the present study (median 4.2 µg/L) were higher. Compared to other countries of the Arctic, the Greenlandic population have high levels of Hg [63,64]. 

Levels of Pb varied nonsignificantly across the studied regions with the highest levels found in East, which may originate from mining activities as well as the high rate of tobacco smoking, as some studies found smoking during pregnancy increased the level of Pb [23,65,66]. Previous studies have found Pb to be associated with low birth weight [26,65]. In the present study, we also observed significantly lower birth weight for the outcome with the highest quartile of blood Pb.

#### 4.2.2. Arsenic and Cadmium

The level of As was significantly different across regions, with Disko Bay and West having the highest median values, which were also above the reported normal range for non-Inuit and nonpregnant [48] (Appendix A). This may be explained by the natural deposits of As or the level of human activities and local pollution within the regions. As has been found to be associated with lower birth weight [26,65,67]. The mechanisms are not fully understood, but it has been suggested that the mechanisms for adverse pregnancy outcomes may relate to the transplacental nature of As and reduced nutrient supply as a consequence of arsenic driven vasoconstriction [68]. Relatively inefficient methylation of As by the mother has also been suggested as a potential cause of low birth weight [69,70]. We found no association between As exposure and birth weight.

Several studies found Cd to be associated with iron deficiency and vice versa, as well as with low birth weight [9,26,65]. Furthermore, in the present study, Cd correlated significantly with cotinine and smoking; smoking reportedly being one of the most important factors for high blood levels of Cd [9,71]. This is also found in other circumpolar populations [29,72,73]. However, in our study, Cd and Fe were weakly positively correlated. This may be a biological response to smoking, which results in more Cd, leading to the requirement of more Fe to bind oxygen in hemoglobin and thus, to maintain oxygen transportation. In the present study, levels of Cd were above reported non-Inuit normal ranges (Appendix A), but nonsignificantly different across regions with the highest levels in the North, South, and East. High levels of Cd may reduce the beneficial effects of some essential nutrients, leading to oxidative stress, DNA damage, mutation, impaired DNA repair, and altered P53 expression [74].

#### 4.2.3. Chromium and Nickel

Blood levels of Cr and Ni were not significantly different across regions but both significantly above the reported normal range in all regions [48,49]. For Ni, it was observed that the East region had higher levels than the remaining four regions. For Cr, the North region had lower levels than the four remaining regions. Calculation of the median blood values of several regions may have been affected by the detection limit described in the limitations of the study. Cr and Ni are important components of stainless steel. Ingestion of drinking water from stainless steel appliances, e.g., faucets, seems a plausible source of Ni and Cr, as well as industrial and agricultural practices and handling of batteries, paints or jewelry [9,29,75].

### 4.3. Blood Levels of Essential Metals

Toxic elements such as Hg, Pb, and Cd may interfere and compete metabolically with essential metals such as Se, Fe, Cu, Zn, and Ca [70,76].

#### 4.3.1. Iron, Zinc, and Magnesium

The levels of Fe were significantly different across regions, with the lowest concentrations in the East region. Fe was significantly inversely correlated with smoking, which was higher in the East.

The ability to lower Fe concentration may be due to the binding of heavy metals on protein sites, displacing the original Fe from their natural binding sites, causing malfunctioning of cells [9,10,11,70]. The WHO recommends 30–60 mg daily Fe supplementation to support the physiologic hemodilution during pregnancy, prevent insufficient supply, anemia, and adverse birth outcomes, e.g., low birth weight and preterm birth [77,78,79].

Regional differences were also observed for Zn, with the highest level in the South and North, and Mg also displayed regional differences with the highest level in the East.

The explanation for the significant regional differences in Zn and Mg blood levels may be human activities such as mining [80,81]. Zn was significantly inversely correlated to smoking. This inverse relationship has previously been reported and may be because smoking is the one of the most important sources of Cd, consequently resulting in higher Cd affecting the zinc levels [71,82,83]. Potential mechanisms are previously suggested and involve competitive zinc displacement by cadmium, e.g., from several DNA repair enzymes [9,10,11,74].

#### 4.3.2. Combinations

A study on an American population suggested that combinations of some essential metals (e.g., Fe, Se, Ca) may mitigate the negative effects of some heavy metals in chronically exposed populations [26]. Furthermore, the study found that higher levels of Cu, Zn, Mg, and Mn may be associated with higher levels of Pb and Cd [26]. The study did not investigate an Arctic population, and the findings require larger longitudinal studies. Literature is scarce regarding the combined toxicity of heavy metals [11].

Nutrients and antioxidants such as Se and the n3/n6 ratio found in seafood are thought to be capable of attenuating the adverse effects of some environmental pollutants [84]. Further research on the relationships between nutrients and contaminants is needed.

A Canadian birth cohort study found that estimated Ca intake may be associated with lower maternal Cd, Pb, Mn, and Hg, as well as cord blood Pb [85]. In support, in the present study, we observed a negative correlation of Ca and Pb, As, Cr, Mn, Zn, as well as Mg. Metal correlation data from the present study are given for all regions in Appendix A, stratified by region in Appendix A. The present study did not examine the effect of mixture exposure.

### 4.4. The Correlations of Metals and Lifestyle

Several metals correlated with lifestyle factors. Age was positively correlated with Pb, Se, and Fe but negatively with Cd and Ca (Table 3). Cd and Pb were positively and As negatively correlated with smoking and cotinine. Notably, smoking and cotinine correlated negatively with the levels of Fe and Zn. Thus, lifestyle can affect the level of metals and have a possible adverse effect on, e.g., fetal growth [23,28,32,34,37,51].

The fatty acids n-3/n-6 ratio correlated significantly positively with plasma-Se and whole blood selenium, respectively. This suggests that the traditional marine diet is an important source of this antioxidant. By contrast, the levels of Hg and Pb were also positively correlated with the n-3/n-6 ratio, suggesting the traditional marine diet was an important source of these heavy metals as well—a classic example of the so-called “Arctic dilemma”, describing the benefits and disadvantages of traditional food within the Arctic regions [1,32,86].

Metal-to-metal correlation analysis showed positive correlation between heavy metals such as Hg, Pb, As, and Cd, suggesting that the contamination from metals in the blood of the women may originate from the same exposure source (Appendix A). Cd may consume or disrupt Se, an antioxidant, since plasma-Se was significantly negatively correlated with Cd.

When stratified by region, these correlations showed some similarities and differences, suggesting different sources of exposure within the regions (Appendix A).

### 4.5. Birth Outcome Characteristics and Association to Prenatal Metal Exposure

No statistically significant differences were found for the birth outcome characteristics across regions, but a tendency for lower birth weight and shorter length was observed in the East. This might be explained by the higher frequency of smokers during pregnancy in the East [65]. In the present study, the East region had the highest heavy metal levels of Hg, Pb, and Ni, second highest of Cd, Cr, and Mn, but the lowest level of As. For the essential metals, the East had the highest levels of Se, Mg, and Ca and the lowest levels of Fe, Cu, and Zn. To elucidate the regional differences, further studies with a higher number of regional participants are needed.

For the total study population, we found that Pb, Cd, and Cu levels were negatively related to birth outcomes. Moreover, the β estimates indicated that an increased level of Cu increases the risk of earlier delivery (Table 6c). Further studies are needed to elucidate how birth outcomes are affected by the exposure to heavy metals and essential metals/trace elements.

### 4.6. Gender Stratification

When stratified by gender, these associations showed clear differences in metal blood levels (Appendix A). For females, Pb and Ni were important risk factors for low birth weight, but Cr correlated positively to birth weight. For girls, head circumference and gestational age were associated with Ni, while for boys, no significant estimates were found after adjustments (Appendix A).

Gender differences were also reported for exposure to other environmental contaminants; e.g., the lipophilic POPs [1]. The heavy metal findings of the present study may provide information for governmental authorities to differentiate prevention programs.

### 4.7. Odds Ratio: Association between Prenatal Exposure to Metals and Risk of Low Birth Weight, Low APGAR5, and Preterm Birth

The OR of 1.441 (95%CI: 1.018; 2.039; *p* = 0.040) of Cd and low birth weight showed a higher risk of low birth weight upon exposure to Cd. Similar findings are previously described in the literature [26]. Levels of Cd are known also to be high in other Inuit populations [87]. A positive association was also found between low birth weight and Ni, indicating that increased levels of Ni increase the risk for low birth weight. This was previously found, although sparsely investigated in literature, but known toxic effects of nickel include hemolysis, genetic alterations, and oxidative stress [88,89,90]. Some findings suggest no association [90].

OR of low APGAR5 and Cd was below 1 (OR = 0.149; 95%CI: 0.028; 0.783; *p* = 0.025) (Table 6b), suggesting that exposure to Cd may reduce the risk of low APGAR5. However, this finding needs to be considered in relation to sensitivity, specificity, and the subjective assessments and is further discussed in the limitations section.

Although weakly, the OR for preterm birth was positively associated with Cu (Table 6c). This association is not well described in the literature. Maternal plasma concentrations of Cu in a Chinese study were found to be lower in preterm births compared to full-term births in umbilical cord blood, although it is stated that this finding requires further studies [91]. For the present study, we found that the birth weight and length decreased by higher quartiles of Cu, but the APGAR score borderline increased (Appendix A). Again, the evaluation of APGAR5 scores must be assessed with some caution.

### 4.8. Strengths and Limitations

The strengths of the present study are the relatively large population compared to previous studies of the Arctic populations. Our results are the first to provide all included population data from the ACCEPT study. The relatively large number of measured metals of this study, compared to the typical analyses in the available literature, provides new insights into heavy metal effects on essential metals, e.g., Fe in the Greenlandic Inuit.

This study has some limitations: Reference ranges of whole blood metal levels are nonexistent for Arctic populations. The reference population for normal ranges included in this study was non-Arctic, nonpregnant Nordic populations both male and female of various ages. No suitable whole blood references for Mg and Ca were found.

Five newborns died within the first 24 hours; we reran the statistical analyses upon eliminating these five, and the statistically significant data given were not changed.

Several non-expected estimates with APGAR5 were found, suggesting that the score improved with higher concentrations of toxic heavy metals, e.g., Cd and Cu, whereas increase in Cr lowered the APGAR5 scores. The effects of heavy metal exposure on the APGAR score is sparsely investigated in the literature, and the conclusions from the few studies that describe the matter differ on population, material, and statistical power and, thus, cannot be generalized to the present study [92,93,94]. Our estimates may reflect that the outcome of APGAR scoring does not take heavy metal levels into account. The adverse effects of Cd, Cr, and Cr may become especially evident with neurological deficits, behavior, and psychomotor function later in the children’s life [9,11,16]. The liability and limitations of the APGAR score have been discussed for decades, e.g., sensitivity, specificity, and the subjective assessments from health care professionals [45,95]. In several studies, neonatologists have found the APGAR score of little value [96,97]. A pilot study compared the neonatal resuscitation and adaption score (NRAS) to the APGAR score, finding the NRAS superior [98]. Maternal Fe supplementation did not benefit the APGAR score in the NRAS study [98].

In a quality control of the ACCEPT analyses, the number of samples eligible for analysis by ICP-MS was investigated. Some analyses of the metals did not exceed the detection limit and, thus, may influence the precision of the present study. Furthermore, 201 samples were not analyzed for Mg and Ca from 2013–2015 as well as 17 samples for Cu (Appendix A).

Adjustments for POPs were not conducted in this study but should be considered for future studies. Adverse health effects associated with POPs in pregnant Inuit women, the fetus, and newborn are found in several studies [2,23,27,30,32,50,99,100]. The Arctic populations have some of the highest levels of POPs globally [1].

When divided into five regions, the study population was relatively small. Consequently, some regions (North, South, East) had below 10% of the total study population included because of the relatively low proportion of residents in these regions, as reported in similar studies, and thus, the statistical power is low [23,28,33,34].

Multiple hypothesis tests with an alpha error set to 0.05 were performed. In the present study, we measured 13 metals in maternal blood, analyzed, and estimated their association with birth outcomes. The ‘multiple testing problem’ should be addressed and chance findings cannot thus be ruled out [101,102].

### 4.9. Diet and Food Insecurity

The Greenlandic Inuit are exposed to high levels of heavy metals through a traditional diet. Previous ACCEPT studies reported a traditional food intake of 12–18% [28,34], a decline from 21% in 2005–2010 [103]. Concerns about changes in diet have previously been reported [1,28,34,51,104,105,106].

It cannot be ruled out that socioeconomic status and food insecurity are factors for the consumption of heavy metal or POP-contaminated food items [107]. In a population survey, about 12% had no food or no money for food [50]. Furthermore, 8% of the 15-year-olds were going to bed hungry “often” or “sometimes” [108]. Food insecurity was reported from current smokers in more than twice the number of nonsmokers [50].

The Arctic is an area of transition, both culturally with regards to lifestyle but also with respect to climate change and pollution. We believe this study elucidates important environmental issues to consider when assessing the human health of the Arctic.

Our results may provide local governments and other stakeholders with information to base their decisions on regarding contaminants and public health. Elucidating the heavy metal effects on the Inuit population and their environment is essential for a complete health risk assessment and management of public health. Hence, continuous monitoring of changes in climate and culture is vital to interpret and understand the complex and synergistic effects between climate change, contaminants, and human health to reduce adverse health effects.

## 5. Conclusions

This cross-sectional ACCEPT study investigated the association between blood metal concentration in Greenlandic pregnant Inuit women and their birth outcomes. High smoking frequency was observed, especially in the East region. For the East region, a tendency for lower birth weight and lower birth length were found compared to the other regions in Greenland. Maternal blood levels of Cd, Cr, and Ni were found to be higher than reported clinical references for nonpregnant, non-Inuit populations (Appendix A). Cd and Cu were associated with reduced birth weight; Cd was associated with reduced birth length; and Zn was weakly associated with reduced head circumference. Highest quartile of Pb had lower birth weight and head circumference. Cu was related to a risk of preterm birth. Cadmium was a significant risk factor for several negative birth outcomes in all regions of this study. Detection of heavy metals in maternal blood can reflect fetal exposure, which may influence fetal development and growth in a gender-dependent dose–response relationship (Appendix A). Lifestyle factors are important sources of toxic heavy metals such as Pb, Cd, and Cr in the Inuit population, and further preventive measures should be considered. Further studies on the health effects of prenatal exposure to heavy metals as well as other environmental contaminants in the Greenlandic Inuit are needed.

## Figures and Tables

**Figure 1 ijerph-16-01171-f001:**
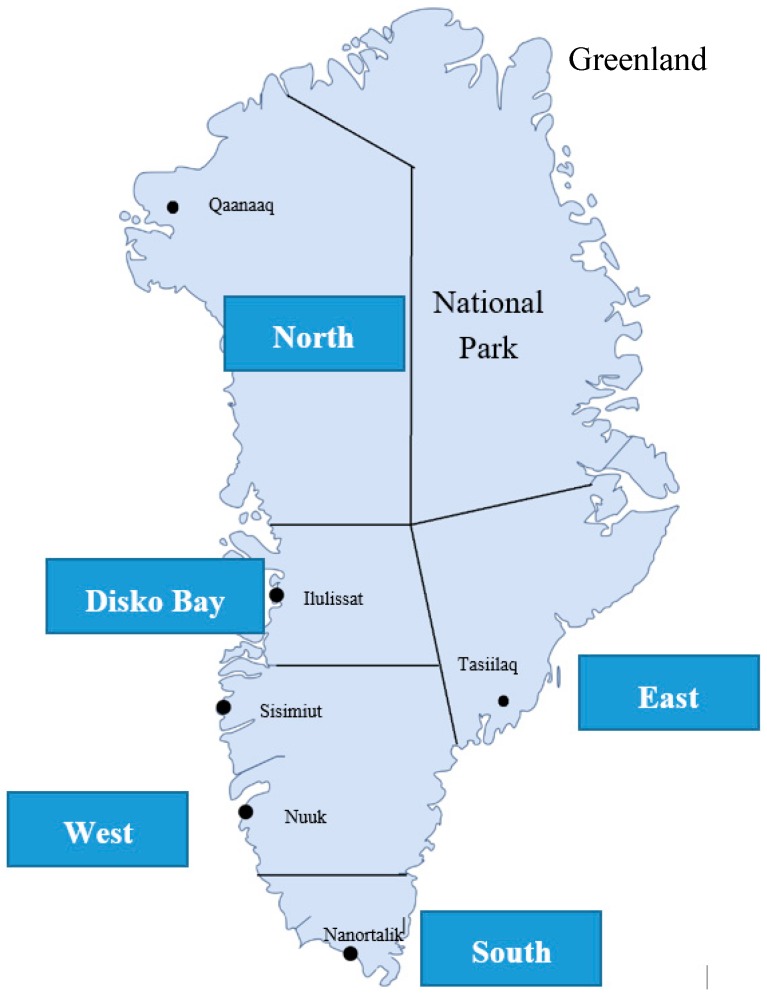
Map of Greenland divided into five regions with indication of residential areas.

**Figure 2 ijerph-16-01171-f002:**
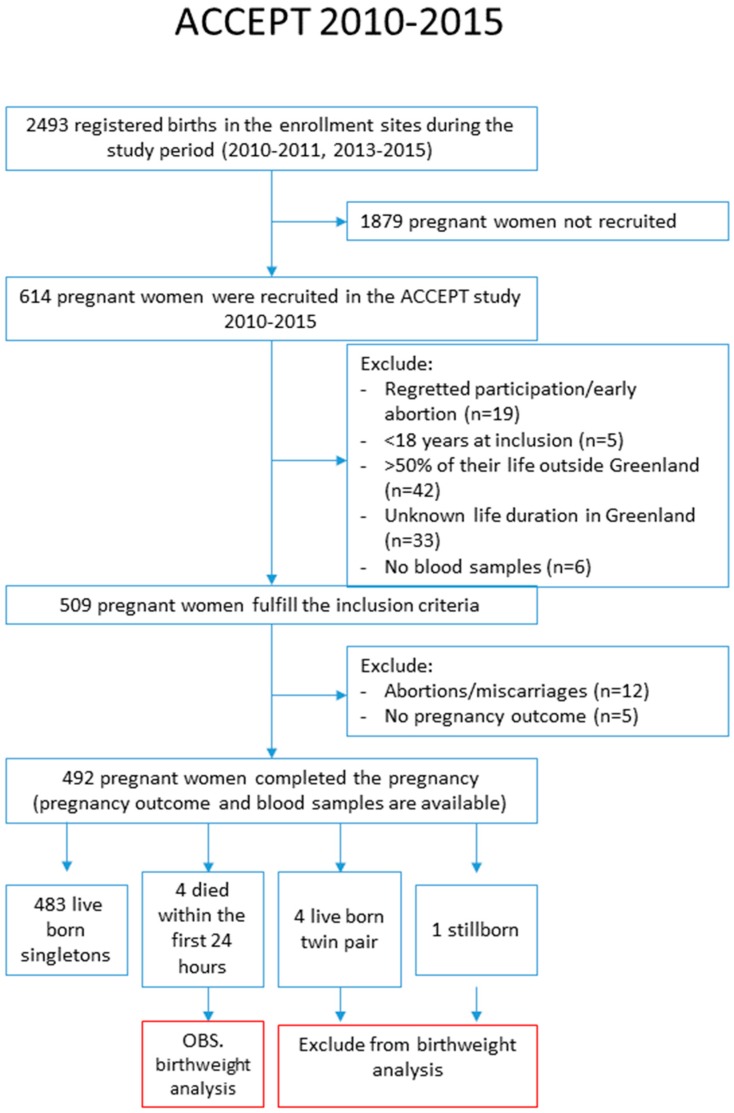
Flowchart of enrolment.

**Table 1 ijerph-16-01171-t001:** Characteristics of the study population: Pregnant women in Greenland.

	Region^1^	North (RN)	Disko Bay (DB)	West (W)	South (S)	East (E)	*p*	Total
N (%)		33 (6.5)	123 (24.2)	289 (56.8)	44 (8.6)	20 (3.9)	0.540#	509 (100)
Age (years)	n	33	123	289	44	20	0.222*	509
	Mean ± SD	28.4 ± 4.3	27.1 ± 5.1	27.4 ± 5.1	28.6 ± 4.1	28.2 ± 6.4		27.5 ± 5.0
	Median (min-max)	27 (20-36)	27 (18-41)	27 (18-42)	29 (20-41)	28 (19-43)		27 (18-43)
BMI^2^	n	31	119	271	43	17	0.403*	481
	Mean ± SD	24.3 ± 3.6	25.7 ± 5.2	25.6 ± 4.9	25.0 ± 4.8	27.1 ± 5.8		25.6 ± 4.96
	Median (min-max)	23.5 (18.7-33.6)	24.7 (17.6-39.4)	24.4 (16.4-46.6)	24.0 (17.5-39.5)	24.8 (19.3-39.0)		24.3 (16.4-46.6)
Education	n	29	115	265	44	20	0.709#	473
	n Primary school (%)	9 (31.0)	39 (33.9)	71 (26.8)	11 (25.0)	9 (45.0)		139 (29.4)
	n Secondary school (%)	13 (44.8)	53 (46.1)	128 (48.3)	21 (47.7)	8 (40.0)		223 (47.1)
	n University (%)	7 (24.1)	23 (20.0)	66 (24.9)	12 (27.3)	3 (15.0)		111 (23.5)
Current smoking^3^	n	32	121	279	44	18	**0.014**#	494
	Yes (%)	13 (40.6)	39 (32.2)	94 (33.7)	18 (40.9)	13 (72.2)		177 (35.8)
Cotinine; *Smokers* (µg/L)	n	13	39	93	18	4	0.441*	167
	Mean ± SD	53.0 ± 53.6	104.7 ± 85.7	69.0 ± 65.7	49.9 ± 45.5	99.5 ± 66.3		74.8 ± 70.2
	Median (min-max)	46.7 (0.5-160.0)	100.3 (0.5-473.0)	53.2 (0.5-297.7)	45.2 (0.5-126.1)	106.9 (13.3-171)		63.7 (0.5-473.0)
Cotinine; *non-smokers* (µg/L)	n	19	82	185	26	3	0.583*	315
	Mean ± SD	23.0 ± 77.1	9.14 ± 25.7	7.7 ± 27.1	12.5 ± 35.0	2.8 ± 4.04		9.4 ± 32.5
	Median (min-max)	0.5 (0.5-337.7)	0.5 (0.5-145.1)	0.5 (0.5-253.8)	0.5 (0.5-170.5)	0.5 (0.5-7.5)		0.5 (0.5-337.7)
Alcohol during pregnancy	n	25	74	180	31	13	0.08#	323
	≤ 1/month (%)	24 (96.0)	74 (100.0)	180 (100)	31 (100)	13 (100)		322 (99.6)
	>2-3/month (%)	1 (5.0)	0 (0.0)	0 (0.0)	0 (0.0)	0 (0.0)		1 (0.3)
Parity	n	29	109	248	34	17	**0.013***	437
	Mean ± SD	1.0 ± 0.9	1.2 ± 1.1	0.9 ± 0.9	0.9 ± 0.8	1.7 ± 1.7		1.0 ± 1.0
	Median (min-max)	1.0 (0-3)	1.0 (0-6)	1.0 (0-5)	1.0 (0-3)	2.0 (0-7)		1.0 (0-7)
Parity grouped	0 (%)	9 (31.0)	30 (27.5)	97 (39.1)	14 (41.2)	4 (23.5)	0.147#	159 (34.7)
	1-2 (%)	13 (44.8)	68 (62.4)	126 (50.8)	17 (50.0)	10 (58.8)		166 (36.2)
	≥ 3 (%)	7 (24.1)	11 (10.1)	25 (10.1)	3 (8.80)	3 (17.6)		133 (29.0)
n-3/n-6^4^	n	33	123	288	44	20	0.277*	508
	Mean ± SD	0.24 ± 0.10	0.24 ± 0.12	0.25 ± 0.13	0.23 ± 0.10	0.29 ± 0.11		0.24 ± 0.12
	Median(min-max)	0.22 (0.11-0.55)	0.21 (0.09-1.20)	0.21 (0.07-0.88)	0.22 (0.09-0.62)	0.26 (0.13-0.57)		0.21 (0.07-1.20)

N: Total number of participants within the region; n: Total number of participants that supplied information for the corresponding parameter; *p* value calculated with (*)One-way ANOVA on ln-transformed data, and (#)Chi^2^ test.; ^1^ The region in which the women lived for > 50% of their lives; ^2^ Body Mass Index, calculated as kg/m^2^; ^3^ Current smoking: The status is obtained from questionnaires/medical journals. The women answered the questions approximately in gestational week 12. The Yes-percentage is calculated from the total answers in the region; ^4^ Ratio between n-3 polyunsaturated fatty acids and n-6 fatty acids.

**Table 2 ijerph-16-01171-t002:** Levels of blood metals in the pregnant women.

Metal (µg/L)	Region^1^	North (RN)	Disko Bay (DB)	West (W)	South (S)	East (E)	*p*	Total
Hg	n	32	121	285	44	20	**<0.001**	502
	Mean ± SD	8.9 ± 9.7	6.4 ± 7.8	4.7 ± 5.6	4.4 ± 3.4	11.5 ± 6.9		5.6 ± 6.6
	Median(min-max)	6.2(1.3-54.5)	4.2(0.8-69.3)	3.2(0.3-73.0)	3.6(0.8-16.8)	10.4(1.5-29.3)		4.2(0.3-73.0)
Pb	n	32	121	285	44	20	0.069	502
	Mean ± SD	8.9 ± 5.5	7.9 ± 5.7	8.6 ± 6.9	9.1 ± 8.4	12.4 ± 12.3		8.6 ± 7.0
	Median(min-max)	7.1(4.3-26.3)	5.3(1.6-31.5)	6.7(1.6-64.1)	6.1(2.7-48.4)	7.0(4.3-58.4)		6.3(1.6-64.1)
As	n	32	121	285	44	20	**0.022**	502
	Mean ± SD	6.2 ± 5.0	8.1 ± 8.2	6.4 ± 4.4	5.5 ± 3.0	4.2 ± 2.6		6.6 ± 5.5
	Median(min-max)	4.2(1.1-26.3)	5.5(0.03-58.8)	4.2(1.1-29.7)	4.2(1.1-18.9)	3.1(2.1-10.4)		4.2(0.03-58.8)
Cd	n	32	121	285	44	20	**0.011**	502
	Mean ± SD	1.2 ± 0.8	1.54 ± 1.6	1.2 ± 0.9	1.2 ± 0.9	1.7 ± 0.6		1.3 ± 1.1
	Median(min-max)	1.6(0.2-4.0)	1.4(0.1-10.8)	1.3(0.1-7.8)	1.1(0.1-4.8)	1.5(1.1-3.7)		1.5(0.1-10.8)
Cr	n	32	121	285	44	20	**0.042**	502
	Mean ± SD	19.8 ± 19.6	34.8 ± 31.5	30.6 ± 43.9	33.2 ± 34.1	19.9 ± 21.4		30.7 ± 38.5
	Median(min-max)	11.2(3.7-72.5)	16.8(3.7-127.8)	16.8(2.5-402.3)	16.8(3.7-143.9)	13.1(3.7-82.4)		16.8(2.5-402.3)
Mn	n	32	121	285	44	20	0.69	502
	Mean ± SD	17.9 ± 7.4	21.1 ± 12.5	20.3 ± 17.1	20.1 ± 14.3	22.3 ± 15.3		20.4 ± 15.3
	Median(min-max)	16.6(7.4-33.6)	17.7(3.4-80.9)	15.7(3.4-236.4)	15.0(3.7-67.2)	17.3(9.6-81.4)		16.0(3.4-236.4)
Ni	n	32	121	285	44	20	0.891	502
	Mean ± SD	14.5 ± 12.8	82.2 ± 670.8	34.5 ± 275.9	15.8 ± 18.1	14.4 ± 5.3		42.3 ± 389.3
	Median(min-max)	12.1(1.6-74.9)	12.1(1.6-7308)	12.1(1.6-4630.5)	12.1(1.6-122.9)	17.9(2.9-17.9)		12.1(1.6-7308)
Se	n	32	121	285	44	20	**0.019**	502
	Mean ± SD	208.0 ± 196.5	186.7 ± 259.7	152.8 ± 189.9	122.1 ± 35.9	150.2 ± 54.8		161.7 ± 199.2
	Median(min-max)	149.9(59.8-1095.9)	124.7(62.0-2248.0)	115.5(45.4-2795.2)	114.6(74.4-258.5)	137.6(83.0-268.8)		120.5(45.4-2795.2)
Plasma-Se	n	31	122	285	44	20	0.785	502
	Mean ± SD	81.1 ± 26.7	72.9 ± 24.4	75.4 ± 27.6	77.1 ± 28.3	74.0 ± 18.6		75.2 ± 26.5
	Median(min-max)	84.0(29.4-157.4)	70.9(27.3-153.3)	75.9(5.3-177.0)	77.0(23.1-144.9)	75.3(37.8-105.0)		75.8(5.3-177.0)
**Metal (mg/L)**								
Fe	n	32	121	285	44	20	**<0.001**	502
	Mean ± SD	432.4 ± 44.7	436.4 ± 71.6	443.6 ± 75.1	492.5 ± 158.0	405.9 ± 144.9		443.9 ± 88.2
	Median(min-max)	443.3(275.7-499.8)	430.8(279.4-928.9)	439.6(215.0-1069.2)	457.1(351.4-1092.1)	382.3(255.8-991.0)		436.8(215.0-1092.1)
Cu #	n	32	121	284	44	7	**0.015**	488
	Mean ± SD	1.4 ± 0.3	1.4 ± 0.3	1.4 ± 0.4	1.3 ± 0.3	1.6 ± 0.5		1.4 ± 0.3
	Median(min-max)	1.3(0.9-2.0)	1.4(0.7-2.1)	1.4(0.6-2.7)	1.3(0.6-2.3)	1.6(0.7-2.3)		1.4(0.6-2.7)
Zn	n	32	121	285	44	20	**<0.001**	502
	Mean ± SD	4.9 ± 0.6	4.5 ± 0.7	4.8 ± 1.0	5.3 ± 1.8	4.5 ± 1.3		4.8 ± 1.0
	Median(min-max)	4.9(3.6-6.0)	4.4(2.7-8.2)	4.7(3.1-12.1)	4.9(3.4-12.9)	4.3(2.8-9.3)		4.6(2.7-12.9)
Mg #	n	20	93	198	27	6	**0.007**	344
	Mean ± SD	36.7 ± 27.7	37.1 ± 4.7	38.0 ± 4.2	39.8 ± 6.2	42.3 ± 6.7		38.0 ± 4.6
	Median(min-max)	36.9(32.2-43.8)	36.7(28.7-58.2)	38.0(30.1-58.3)	38.0(31.8-55.0)	41.6(34.7-54.3)		37.6(28.7-58.3)
Ca #	n	20	93	198	27	6	**0.003**	344
	Mean ± SD	64.7 ± 4.6	63.7 ± 9.4	64.2 ± 10.5	56.5 ± 20.3	57.8 ± 25.4		63.4 ± 11.6
	Median(min-max)	65.2(57.3-73.2)	64.3(8.2-84.0)	63.8(5.6-95.0)	62.4(5.0-76.6)	65.2(7.8-76.3)		63.9(5.0-95.0)

n: Total number of participants that supplied information for the corresponding parameter; *p* value calculated by One-way ANOVA on ln-transformed data; ^1^: The region the women lived for > 50% of their lives; #: Calculated with less data, see detection limits in Appendix A.

**Table 3 ijerph-16-01171-t003:** Spearmans correlation coefficient (r_s_) between the blood metals and lifestyle factors of Greenlandic pregnant women.

Metal	Age (years)			Smoking^1^			Cotinine (µg/L)			n-3/n-6^2^		
	n	r_s_	*p*	n	r_s_	*p*	n	r_s_	*p*	n	r_s_	*p*
Hg	502	0.06	0.18	487	0.06	0.18	488	0.07	0.122	501	0.24	**<0.001**
Pb	502	0.10	**0.02**	487	0.06	0.21	488	0.16	**0.001**	501	0.13	**0.004**
As	502	0.11	0.01	487	*−0.10*	**0.03**	488	0.02	0.96	501	*0.02*	0.632
Cd	502	*−0.07*	0.11	487	0.27	**<0.001**	488	0.36	**<0.001**	501	*−0.08*	0.09
Cr	502	0.03	0.47	487	*−0.11*	**0.02**	488	*−0.06*	0.18	501	*−0.14*	**0.002**
Mn	502	*−0.004*	0.93	487	*0.01*	0.84	488	*−0.03*	0.45	501	*−0.15*	**0.001**
Ni	502	0.03	0.50	487	*−0.02*	0.70	488	*−0.06*	0.20	501	0.11	**0.02**
Se	502	0.12	**0.01**	487	*−0.03*	0.45	488	0.01	0.81	501	0.37	**<0.001**
Plasma-Se	502	0.11	**0.01**	487	*−0.08*	0.09	488	*−0.06*	0.22	501	0.21	**<0.001**
Fe	502	0.10	**0.02**	487	*−0.18*	**<0.001**	488	*−0.19*	**<0.001**	501	0.02	0.73
Cu	488	*−0.02*	0.73	475	0.12	**0.01**	488	0.05	0.25	487	−0.06	0.17
Zn	502	0.00	0.99	487	*−0.20*	**<0.001**	488	*−0.20*	**<0.001**	501	*−0.05*	0.30
Mg	344	*−0.003*	0.96	338	*−0.03*	0.57	344	*−0.06*	0.34	344	*−0.02*	0.71
Ca	344	*−0.13*	**0.02**	338	0.16	**0.004**	344	0.13	**0.02**	344	*−0.03*	0.63

n: Total number of participants that supplied information for the corresponding parameter; r_s_: Spearman’s correlation coefficient; *p* values in **bold** highlights statistically significant correlation; *italic* highlights negative correlation; ^1^ The smoking status was obtained from questionnaires/medical journals; ^2^: Ratio between n-3 polyunsaturated fatty acids and n-6 fatty acids, indicator of marine food intake.

**Table 4 ijerph-16-01171-t004:** Birth outcomes of the Greenlandic study population.

Outcome	Region^1^	North (RN)	Disko Bay (DB)	West (W)	South (S)	East (E)	*p*	Total
Birth weight (grams)	n	31	116	277	44	19	0.322*	487
	Mean ± SD	3356.9 ± 614.0	3585.0 ± 613.6	3594.0 ± 578.1	3615.1 ± 490.7	3421.9 ± 320.8		3572.0 ± 575.8
	Median(min-max)	3455.0(1383.0-4330.0)	3640.0(1575.0-5315.0)	3615.0(585.0-5170.0)	3572.5(2642.0-4850.0)	3360.0(2775.0-3975.0		3615.5(585.0-5315.0)
Birth length (cm)	n	31	116	277	44	19	0.471*	487
	Mean ± SD	50.3 ± 2.6	51.3 ± 2.7	51.4 ± 3.1	51.4 ± 2.4	50.6 ± 1.8		51.3 ± 2.9
	Median(min-max)	50.0(41.0-54.0)	51.0(42.0-58.0)	51.0(24.0-58.0)	51.0(46.0-57.0)	50.0(48.0-54.0)		51.0(24.0-58.0)
Head circumference (cm)	n	31	114	277	44	19	0.073*	485
	Mean ± SD	33.8 ± 1.9	34.7 ± 1.6	34.7 ± 1.8	34.6 ± 1.3	34.1 ± 0.7		34.6 ± 1.7
	Median(min-max)	34.0(31.0-37.0)	35.0(30.0-38.5)	35.0(20.5-38.5)	34.7(32.0-37.0)	34.0(33.0-35.0)		35.0(20.5-38.5)
Gestation age (weeks)	n	30	113	269	43	15	0.423*	470
	Mean ± SD	38.5 ± 2.1	39.0 ± 1.6	39.2 ± 2.1	39.3 ± 1.4	38.8 ± 1.7		39.1 ± 1.9
	Median(min-max)	39.0(32.0-41.0)	39.0(33.0-42.0)	40.0(22.0-42.0)	39.0(36.0-42.0)	39.0(36.0-42.0)		39.0(22.0-42.0)
APGAR5	n	31	113	275	44	18	0.795*	481
	Mean ± SD	9.87 ± 0.6	9.82 ± 0.72	9.76 ± 1.02	9.86 ± 0.55	9.89 ± 0.32		9.79 ± 0.88
	Median(min-max)	10(7-10)	10(5-10)	10(1-10)	10(7-10)	10(9-10)		10(1-10)
APGAR5, grouped	n	31	113	275	44	18	0.960#	470
	Score <7 (%)	1 (3.2)	3 (2.7)	6 (2.2)	1 (2.3)	0 (0)		13 (2.6)
	Score >7 (%)	30 (96.8)	110 (97.3)	269 (97.8)	43 (97.7)	18 (100)		495 (97.4)

n: Total number of participants that supplied information for the corresponding parameter; *p* value calculated by (*)One-way ANOVA on ln-transformed data and (#)Chi^2^ test; ^1^:The region the women lived in for > 50% of their lives.

**Table 5 ijerph-16-01171-t005:** Association between prenatal exposure to metals and ***birth weight*** for both genders.

	Raw Data			Adjusted^1^			Adjusted^2^			Adjusted^3^		
Metal (µg/L)	n	β (95%CI)	*p*	n	β (95%CI)	*p*	n	β (95%CI)	*p*	n	β (95%CI)	*p*
Hg	481	−5.127 (−12.889;2.635)	0.195	265	−4.946 (−14.821;4.929)	0.325	255	−0.238 (−8.261;7.911)	0.954	265	−4.336 (−14.185;5.513)	0.387
Pb	481	−6.749 (−14.199; 0.701)	0.076	265	−4.130 (−14.689;6.430)	0.442	255	−2.061 (−10.739;6.616)	0.640	265	−5.320 (−15.881;5.241)	0.322
As	481	1.147 (−8.654;10.947)	0.818	265	8.069 (−3.862;20.001)	0.184	255	5.029 (−5.271;15.328)	0.337	265	8.531 (−3.342;20.405)	0.158
Cd	481	**−75.506 (−122.825;−28.187)**	**0.002**	265	**−78.490 (−136.062;−20.918)**	**0.008**	255	**−55.151 (−102.810;−7.491)**	**0.024**	265	**−77.258 (−134.562;−19.955)**	**0.008**
Cr	481	0.310 (−1.035;1.655)	0.651	265	1.554 (−0.375;3.483)	0.114	255	0.260 (−1.351;1.870)	0.751	265	1.468 (−0.455;3.390)	0.134
Mn	481	1.562 (−1.823;4.947)	0.365	265	2.180 (−1.638;5.998)	0.262	255	0.198 (−2.996;3.392)	0.903	265	1.912 (−1.899;5.723)	0.324
Ni	481	−0.088 (−0.219;0.042)	0.184	265	−0.100 (−0.224;0.024)	0.113	255	−0.059 (−0.160;0.041)	0.246	265	−0.094 (−0.218;0.029)	0.134
Se	481	−0.058 (−0.326;0.210)	0.670	265	0.033 (−0.249;0.314)	0.820	255	0.070 (−0.158;0.298)	0.546	265	0.035 (−0.245;0.316)	0.804
P-Se	481	−0.153 (−2.142;1.835)	0.880	265	−1.358 (−3.686;0.969)	0.252	255	−0.824 (−2.755;1.106)	0.401	265	−1.427 (−3.741;0.887)	0.226
Fe	481	0.0004 (−0.0002;0.0010)	0.213	265	6.53 × 10^−5^ (−0.001;0.001)	0.844	255	−0.0001(−0.001;0.0005)	0.697	265	−9.15 × 10^−5^ (−0.001;0.001)	0.805
Cu	468	**−0.248 (−0.403;−0.093)**	**0.002**	265	**−0.242 (−0.432;−0.051)**	**0.013**	255	−0.088(−0.248;0.073)	0.283	265	**−0.242 (−0.432;−0.053)**	**0.012**
Zn	481	0.033 (−0.016;0.083)	0.187	265	0.010 (−0.051;0.072)	0.743	255	−0.015(−0.067;0.037)	0.574	265	−0.0004 (−0.063;0.062)	0.989
Mg	329	0.008 (−0.005;0.022)	0.234	219	0.005 (−0.011;0.021)	0.568	209	0.005 (−0.013;0.014)	0.387	219	0.002 (−0.014;0.019)	0.774
Ca	329	**−0.0060 (−0.011;−0.001)**	**0.029**	219	−0.006 (−0.012;0.000)	0.052	209	−0.003 (−0.008;0.003)	0.351	219	−0.006 (−0.012;0.0005)	0.072

^1^:Core adjustment: Age, BMI, alcohol during pregnancy, cotinine, parity, n-3/n-6 ratio; ^2^:Core and gestation age adjustment; ^3^:Core and region adjustment.

**Table ijerph-16-01171-t006a:** (**a**)

	Raw Data			Adjusted^1^			Adjusted^2^			Adjusted^3^		
Metal (µg/L)	n	OR (95%CI)	*p*	n	OR (95%CI)	*p*	n	OR (95%CI)	*p*	n	OR (95%CI)	*p*
Hg	481	1.023 (0.977;1.072)	0.334	265	1.031 (0.970;1.097)	0.323	255	1.028 (0.947;1.116)	0.511	265	1.031 (0.959;1.108)	0.407
Pb	481	1.026 (0.976;1.078)	0.314	265	1.048 (0.917;1.244)	0.399	255	1.051 (0.876;1.241)	0.639	265	1.063 (0.990;1.143)	0.094
As	481	1.009 (0.930; 1.094)	0.829	265	0.815 (0.530;1.254)	0.352	255	0.526 (0.183;1.511)	0.233	265	0.827 (0.543;1.259)	0.376
Cd	481	1.202 (0.898;1.610)	0.216	265	**1.441 (1.018;2.039)**	**0.040**	255	**1.517 (1.022;2.252)**	**0.039**	265	1.425 (1.005;2.019)	**0.047**
Cr	481	1.005 (0.997;1.013)	0.201	265	0.993 (0.958;1.261)	0.690	255	0.994 (0.941;1.051)	0.839	265	0.995 (0.958;1.033)	0.790
Mn	481	0.996 (0.961;1.032)	0.837	265	0.928 (0.827;1.042)	0.205	255	0.943 (0.826;1.078)	0.390	265	0.921 (0.812;1.046)	0.207
Ni	481	**1.001 (1.00009;1.001103)**	**0.021**	265	**1.001 (1.00003;1.001)**	**0.040**	255	1.001 (0.999;1.002)	0.314	265	1.001 (0.99988;1.0013)	0.095
Se	481	1.001 (0.999;1.002)	0.404	265	1.001 (0.999;1.002)	0.617	255	1.001 (0.998;1.003)	0.555	265	1.001 (0.999;1.003)	0.511
P-Se	481	1.001 (0.982;1.018)	0.992	265	0.995 (0.970;1.021)	0.763	255	0.988 (0.956;1.021)	0.483	265	0.993 (0.963;1.023)	0.631
Fe	481	**0.99999 (0.99998;0.99999)**	**0.020**	265	0.999998 (0.999986;1.000009)	0.703	255	1.000002 (0.999992;1.000012)	0.767	265	1.000 (0.99999;1.00001)	0.954
Cu	468	1.001 (0.999;1.002)	0.348	265	1.000 (0.997;1.002)	0.670	255	0.998 (0.996;1.001)	0.229	265	0.999 (0.997;1.002)	0.882
Zn	481	0.999480 (0.999;1.0002)	0.132	265	0.9998 (0.999;1.0007)	0.623	255	1.0002 (0.9993;1.0011)	0.685	265	0.9993 (0.9984;1.0003)	0.167
Mg	329	0.9999 (0.9997;1.00002)	0.075	219	0.99994 (0.99971;1.0002)	0.593	209	0.99995 (0.99970;1.00022)	0.725	219	0.9998 (0.9987;1.0009)	0.755
Ca	329	1.00004 (0.99997;1.00011)	0.247	219	0.99999 (0.99992;1.00006)	0.831	209	0.99997 (0.99989;1.00006)	0.568	219	0.99996 (0.99969;1.0002)	0.614

^1^:Core adjustment: Age, BMI, alcohol during pregnancy, cotinine, parity, n-3/n-6 ratio; ^2^:Core and gestation age adjustment; ^3^:Core and region adjustment.

**Table ijerph-16-01171-t006b:** (**b**)

	Raw Data			Adjusted^1^			Adjusted^2^			Adjusted^3^		
Metal (µg/L)	n	OR (95%CI)	*p*	n	OR (95%CI)	*p*	n	OR (95%CI)	*p*	n	OR (95%CI)	*p*
Hg	475	0.936 (0.790;1.108)	0.442	262	0.807 (0.574;1.133)	0.215	253	0.803 (0.575;1.132)	0.211	262	0.755 (0.502;1.135)	0.177
Pb	475	0.955 (0.836;1.091)	0.500	262	0.953 (0.792;1.149)	0.616	253	0.947 (0.784;1.143)	0.173	262	0.958 (0.791;1.160)	0.659
As	475	1.027 (0.945;1.116)	0.526	262	1.048 (0.934;1.177)	0.425	253	1.053 (0.939;1.182)	0.356	262	1.047 (0.934;1.173)	0.430
Cd	475	**0.328 (0.111;0.971)**	**0.044**	262	**0.149 (0.028;0.783)**	**0.025**	253	**0.148 (0.028;0.777)**	**0.024**	262	**0.151 (0.029;0.803)**	**0.027**
Cr	475	1.006 (0.997;1.015)	0.202	262	1.006 (0.992;1.019)	0.402	253	1.007 (0.994;1.020)	0.305	262	1.006 (0.993;1.020)	0.338
Mn	475	0.997 (0.953;1.042)	0.886	262	0.979 (0.901;1.065)	0.622	253	0.983 (0.903;1.070)	0.695	262	0.983 (0.901;1.072)	0.692
Ni	475	0.997 (0.975;1.020)	0.820	262	0.995 (0.962;1.028)	0.760	253	0.995 (0.958;1.033)	0.781	262	0.995 (0.959;1.032)	0.777
Se	475	0.997 (0.987;1.007)	0.505	262	0.990 (0.969;1.010)	0.325	253	0.989 (0.968;1.011)	0.316	262	0.989 (0.968;1.010)	0.315
P-Se	475	1.002 (0.980;1.025)	0.859	262	1.002 (0.972;1.032)	0.915	253	1.002 (0.971;1.033)	0.921	262	1.002 0.971;1.034)	0.915
Fe	475	0.999997 (0.999987;1.000007)	0.553	262	0.999980 (0.999956;1.0000)	0.103	253	0.99998 (0.999956;1.00000)	0.101	262	0.99998 (0.999957;1.00000)	0.113
Cu	462	0.999 (0.997;1.001)	0.181	262	0.999 (0.996;1.001)	0.315	253	0.999 (0.996;1.001)	0.359	262	0.999 (0.996;1.002)	0.377
Zn	475	0.9997 (0.9989;1.0005)	0.399	262	0.999 (0.998;1.001)	0.473	253	0.9995 (0.998;1.001)	0.459	262	0.999 (0.998;1.001)	0.512
Mg	325	0.99995 (0.99975;1.00014)	0.576	217	0.9999 (0.9995;1.0002)	0.456	208	0.9999 (0.999;1.000)	0.455	217	0.99985 (0.999;1.0002)	0.446
Ca	343	1.00000 (0.99994;1.00006)	0.992	217	0.99998 (0.99988;1.00009)	0.765	208	0.99999 (0.99988; 1.00010)	0.826	217	0.99999 (0.99988;1.00010)	0.842

^1^:Core adjustment: Age, BMI, alcohol during pregnancy, cotinine, parity, n-3/n-6 ratio; ^2^:Core and gestation age adjustment; ^3^:Core and region adjustment.

**Table ijerph-16-01171-t006c:** (**c**)

	Raw data			Adjusted^1^			Adjusted^2^		
Metal (µg/L)	n	OR (95%CI)	*p*	n	OR (95%CI)	*p*	n	OR (95%CI)	*p*
Hg	464	1.018 (0.986;1.051)	0.265	255	1.019 (0.975;1.064)	0.401	255	1.017 (0.973;1.063)	0.462
Pb	464	1.011 (0.975;1.048)	0.566	255	1.003 (0.940;1.070)	0.933	255	1.007 (0.908;1.070)	0.823
As	464	1.002 (0.953;1.053)	0.943	255	0.974 (0.881;1.077)	0.609	255	0.974 (0.882;1.076)	0.607
Cd	464	1.209 (0.996;1.468)	0.055	255	1.150 (0.876;1.509)	0.315	255	1.143 (0.870;1.501)	0.337
Cr	464	0.997 (0.989;1.005)	0.443	255	0.990(0.972;1.009)	0.319	255	0.990 (0.971;1.010)	0.325
Mn	464	0.983 (0.958;1.008)	0.174	255	0.967 (0.922;1.015)	0.173	255	0.967 (0.921;1.016)	0.182
Ni	464	1.0003 (0.9999;1.0008)	0.149	255	1.0003 (0.9999;1.001)	0.174	255	1.0003 (0.9998;1.0008)	0.212
Se	464	1.001 (0.999;1.002)	0.333	255	1.000 (0.999;1.002)	0.668	255	1.000 (0.999;1.002)	0.661
P-Se	464	1.006 (0.996;1.016)	0.226	255	1.006 (0.994;1.018)	0.335	255	1.007 (0.994;1.019)	0.307
Fe	464	0.999996 (0.999991;1.000000)	0.092	255	0.999996 (0.99999;1.000003)	0.286	255	0.999997 (0.99999;1.000004)	0.355
Cu	454	**1.001 (1.0005; 1.002)**	**0.002**	255	**1.001 (1.0001;1.002)**	**0.031**	255	**1.001 (1.0002;1.003)**	**0.026**
Zn	464	**0.9996 (0.9992;0.9999)**	**0.032**	255	0.9998 (0.9994;1.0003)	0.450	255	0.9999 (0.9993;1.0003)	0.535
Mg	316	**0.999900 (0.999817; 0.999983)**	**0.018**	209	0.99992 (0.9998;1.00002)	0.129	209	0.999927 (0.999817;1.000038)	0.198
Ca	316	**1.00005 (1.00002;1.00009)**	**0.006**	209	1.000046 (0.999997;1.000094)	0.065	209	1.00005 (0.99999;1.000101)	0.057

^1^:Core adjustment: Age, BMI, alcohol during pregnancy, cotinine, parity, n-3/n-6 ratio; ^2^:Core and region adjustment.

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
