# Peer review of "Pregnant Inuit Women’s Exposure to Metals and Association with Fetal Growth Outcomes: ACCEPT 2010–2015"

_ijerph, 2019, doi:10.3390/ijerph16071171_

Round 1

Reviewer 1 Report

General comments:
The manuscript "Pregnant Inuit women’s exposure to metals and association with fetal growth outcomes: ACCEPT 2010–2015 " presents important results from the cross-sectional study investigated the association between blood metal concentration in Greenlandic pregnant Inuit women and their birth outcomes.

The methods of this paper have been used in several studies in the research group, and here applied to investigate pregnant Inuit women’s exposure to metals. The associations showed maternal blood levels of Cd, Cr and Ni were higher than reference values. Cd and Cu associated with reduced birth weight and length. Cd was a significant risk factor for several negative birth outcomes.

The work has been performed with relevant methods and the results are well presented. The conclusions follow from the results. The text has been carefully written.

Small remarks:
- It would be valuable to mention birth cohort ACCEPT (Adaption to Climate Change, Environmental Pollution, and dietary Transition) full name in the Abstract

Author Response

Author response: Thank you very much for your time and valuable comments. We hope that the revision are satisfactory for you. We made revisions to the number of tables based on one reviewers request, and revised estimates, which is highlighted by the “Track changes” function in the tables and related text.

Point 1
Small remarks: 
- It would be valuable to mention birth cohort ACCEPT (Adaption to Climate Change, Environmental Pollution, and dietary Transition) full name in the Abstract
Response 1: Written at page 1 line 15-16. “This cross-sectional study is part of the ACCEPT birth cohort (Adaption to Climate Change, Environmental Pollution, and dietary Transition) and […]”

Reviewer 2 Report

The authors presented an important study about the health risk of pregnant Inuit women, which was based on the investigation of the relationship between the exposures to heavy metals and the fetal growth outcomes. The data of 539 pregnant women was collected during over five years in the Arctic regions, where few health risk studies have been done. The cross-sectional study was performed by characterizing the maternal and infant medical records and analyzing 13 heavy metals in the blood samples, which was associated with some advanced statistical analysis. The results indicated that there were significant regional differences for metals, smoking habit, and parity, and Ca, Pb and Cu were significantly inverse to birth outcomes. Then, the study concluded that the high frequency of smokers in early pregnancy is a great of concern. This study is very important and highly related to the topic of the journal, of which the results can help local governments better manage heavy metal contaminations and protect public health. It is suggested that this manuscript should be considered for publication with some revisions. 

More conclusions should be added in the abstract

There are 10 keywords in this section. Kindly reduce the number down to 5-6. For example: Greenlandic Inuit; Arctic contaminants; heavy metals; perinatal risks; smoking.

Line 30-31 Some references about the sources, occurrence and pathways of Arctic contaminants should be included: Barrie et al. (1992) (https://doi.org/10.1016/0048-9697(92)90245-N), Hung et al. (2016) (https://doi.org/10.1016/j.envpol.2016.01.079), Bossi et al. (2015) (https://doi.org/10.1016/j.envpol.2015.12.026), etc.

Line 32-33 Kindly provide references about how metal and pop contaminants were introduced to the environment, e.g., Zietz et al. (2008) (https://doi.org/10.1016/j.ijheh.2008.04.001), Jennings et al. (2015) (https://doi.org/10.1016/j.jenvman.2015.06.001), and (https://doi.org/10.1016/j.jenvman.2015.06.001), and how they bioaccumulate through the food web, e.g., Ikemoto et al. (2008) (https://doi.org/10.1016/j.chemosphere.2008.01.011), Gandhi et al. (2006) (10.1021/es052064l).

Line 49-51 The toxicity of contaminants also depends on the exposure dose, exposure route, single or mixtures of the compounds, and physiologically based toxicokinetic (PBTK) model.

Line 68 references needed: Walker et al. (2003) (https://doi.org/10.1016/S0048-9697(02)00319-4), Hansen (2000) (https://doi.org/10.1016/S0378-4274(99)00203-9).

The significance and novelty of the study should be addressed more.

There are so many tables, some of which could be put in in the supporting materials.

Line 511 what are their traditional diets, e.g., fish? Does the food contain high level of metals?

What’s the health implementation and management regarding the results?

Author Response

Author response: Thank you very much for your time and valuable comments. As you can see, we mostly agree. We hope that the revision are satisfactory for you. We made revisions to the number of tables based on your request, and revised estimates, which is highlighted by the “Track changes” function in the tables and related text.

Point 1
More conclusions should be added in the abstract
Response 1: Abstract rewritten page 1 line 23-26: “Heavy metals in maternal blood can adversely influence fetal development and growth in a dose-response relationship. Diet and lifestyle factors are important sources of toxic heavy metals and deviant levels of essential metals.”

Point 2
There are 10 keywords in this section. Kindly reduce the number down to 5-6. For example: Greenlandic Inuit; Arctic contaminants; heavy metals; perinatal risks; smoking.
Response 2: We have revised and reduced the keywords based on your comment. However, the manuscript conforms to the manuscript template/guideline, in which up to ten keywords are allowed. For search engine and indexing purposes the number of keywords left in are 7: Greenland; Arctic; heavy metals; perinatal risks; smoking; reproductive health; environmental pollutants

Point 3
Line 30-31 Some references about the sources, occurrence and pathways of Arctic contaminants should be included: Barrie et al.
(1992) (https://doi.org/10.1016/0048-9697(92)90245-N), Hung et al. (2016) (https://doi.org/10.1016/j.envpol.2016.01.079), Bossi et al. (2015) (https://doi.org/10.1016/j.envpol.2015.12.026), etc.
Response 3: We added the suggested references. Now line 33-34
Barrie et al: Reference # 3
Hung et al: Reference # 4
Bossi et al: Reference # 5
In addition for the revised version: Kirk et al (#6), Ma et al (#7) and Jennings et al (#8).

Point 4
Line 32-33 Kindly provide references about how metal and pop contaminants were introduced to the environment, e.g., Zietz et al. (2008) (https://doi.org/10.1016/j.ijheh.2008.04.001), Jennings et al. (2015) (https://doi.org/10.1016/j.jenvman.2015.06.001), and (https://doi.org/10.1016/j.jenvman.2015.06.001), and how they bioaccumulate through the food web, e.g., Ikemoto et al. (2008) (https://doi.org/10.1016/j.chemosphere.2008.01.011), Gandhi et al. (2006) (10.1021/es052064l).
Response 4: We added the suggested references Now line 34-36

Jennings et al: Reference #8
Ikemoto et al: Reference #13
Gandhi et al: Reference #14
Zietz et al: Reference # 15
In addition for the revised version: Barrie et al (#3), Kirk et al (#6), Hansen (#12)

Point 5
Line 49-51 The toxicity of contaminants also depends on the exposure dose, exposure route, single or mixtures of the compounds, and physiologically based toxicokinetic (PBTK) model.
Response 5: Agree. Re-written. Page 2 line 52-55: “The severity of toxicity as well as organ systems affected depends on the heavy metal type, chemical form, whether it is single or mixtures of compounds, exposure dose and time, exposure route, physiologically based pharmacokinetic model and the age of the exposed organisms including humans”

Point 6
Line 68 references needed: Walker et al.
(2003) (https://doi.org/10.1016/S0048-9697(02)00319-4), Hansen (2000) (https://doi.org/10.1016/S0378-4274(99)00203-9).
Response 6: Added references to page 2, line 74-76. “There are reports on a decreasing trend in blood levels of contaminants for some of the Arctic populations, indicating regulation through legislation as well as health interventions are working”
Walker et al: Reference #35
Hansen: Reference #12
In addition for the revised version: Gibson et al (#37)

Point 7
The significance and novelty of the study should be addressed more.
Response 7: Added page 13 line 481-484: “Our results are the first to provide all included population data from the ACCEPT study. The relatively large number of measured metals of this study, compared to the typical analyses in the available literature, provides new insights of heavy metal effects on essential metals e.g. Fe in the Greenlandic Inuit.”

Point 8
There are so many tables, some of which could be put in in the supporting materials.
Response 8: Table 6a-6b moved to supplementary and renamed Table S20a-S20b.

Point 9
Line 511 what are their traditional diets, e.g., fish? Does the food contain high level of metals?
Response 9: Already accounted for at page 2 line 69-71: “The traditional diet of the Greenlandic Inuit rely on marine food such as seal, whale and polar bears at the top of the food chain and furthermore fish and therefore, they are exposed to relatively higher levels of heavy metals”

Point 10
What’s the health implementation and management regarding the results?
Response 10: Re-written page 12 line 530-538
The Arctic is an area of transition. Both culturally with regards to lifestyle but also with respect to climate change and pollution. We believe this study elucidates important environmental issues to consider when assessing human health of the Arctic.
Our results may provide local governments and other stakeholders with information to base their decisions upon regarding contaminants and public health. Elucidating the heavy metal effects on the Inuit population and their environment is essential for a complete health risk assessment and management of public health. Hence, continuous monitoring of changes in climate and culture is vital to interpret and understand the complex and synergistic effects between climate change, contaminants and human health to reduce adverse health effects.”

Reviewer 3 Report

This manuscript reports the association blood metal concentrations and foetal growth in Greenland Inuit population. 

It presents very interesting results. I believe the following comments will help authors further improve their manuscript before considered for publication.

1) Please explain how authors managed possible environmental and device-related contaminations to blood measurements. Any travel blanks were prepared and analysed?

2) How did authors examine the effect of mixture exposure? Authors measured numbers of metal elements and build the logistic models one element by one? Please explain.

3) Authors did multiple hypothesis tests with an alpha error set to 0.05. Please add discussion about multiplicity problem.

e.g.: https://www.ncbi.nlm.nih.gov/pubmed/21154895

4) Authors mention about POPs in the introduction section but nowhere else. I would recommend authors to limit the introduction to metals.

Author Response

Author response: Thank you very much for your time and valuable comments. As you can see, we mostly agree. We hope that the revision are satisfactory for you. We made revisions to the number of tables based on one reviewers request, and revised estimates, which is highlighted by the “Track changes” function in the tables and related text.

1) Please explain how authors managed possible environmental and device-related contaminations to blood measurements. Any travel blanks were prepared and analysed?
Response 1: Of cause, contamination free Supelco® and Nunc™ tubes, respectively were used for sampling, and in the analyzing laboratory parallel blanks were analyzed.  

Moreover at line 119-122, “The quality was ensured by repeated analyses and frequent analysis of certified reference material (Seronorm), as well as by participation in the Quality Assurance of Information in Marine Environmental monitoring (QUASIMEME), an inter-laboratory comparison programme”.

2) How did authors examine the effect of mixture exposure? Authors measured numbers of metal elements and build the logistic models one element by one? Please explain.
Response 2: We did not analyze mixture effects in our study but data of single compounds only. Thus our models build one element by one.

line 167-168: “We built the models one element by one”

and

line 419-420: “The present study did not examine the effect of mixture exposure.”

3) Authors did multiple hypothesis tests with an alpha error set to 0.05. Please add discussion about multiplicity problem. e.g.: https://www.ncbi.nlm.nih.gov/pubmed/21154895
Response 3: page 11 line 516-519 added following text:

“Multiple hypothesis tests with an alpha error set to 0.05 were performed. In the present study, we measured 13 metals in maternal blood and analyzed and estimated their association with birth outcomes. The ‘multiple testing problem’ should be addressed and chance finding thus cannot be ruled out.”

4) Authors mention about POPs in the introduction section but nowhere else. I would recommend authors to limit the introduction to metals.
Response 4: We partly agree– but another reviewer want us to keep it and for some readers it might be valuable information to read about the POP transport to the Arctic. Therefore, we kept the sentences about POP in the Introduction: Line 33 – 36: “Environmental contaminants such as toxic heavy metals and persistent organic pollutants (POPs) are transported to the Arctic through long-range atmospheric and ocean currents [1–7]. Heavy metals (e.g. mercury, lead and cadmium) and POPs are introduced to the ecosystems where they bio-accumulate and bio-magnify up through the food chain in e.g. fish and marine mammals.

Line 77: Humans are still being exposed to heavy metals as well as POPs.

However, we do point out at the end of the Introduction, that:

Line 77-79: The aim of this study was to investigate the association of Greenlandic pregnant Inuit women's levels of heavy metals and essential metals and the birth outcomes.
